# Sevanol and Its Analogues: Chemical Synthesis, Biological Effects and Molecular Docking

**DOI:** 10.3390/ph13080163

**Published:** 2020-07-24

**Authors:** Olga A. Belozerova, Dmitry I. Osmakov, Andrey Vladimirov, Sergey G. Koshelev, Anton O. Chugunov, Yaroslav A. Andreev, Victor A. Palikov, Yulia A. Palikova, Elvira R. Shaykhutdinova, Artem N. Gvozd, Igor A. Dyachenko, Roman G. Efremov, Vadim S. Kublitski, Sergey A. Kozlov

**Affiliations:** 1Shemyakin-Ovchinnikov Institute of Bioorganic Chemistry, Russian Academy of Sciences, 117997 Moscow, Russia; osmadim@gmail.com (D.I.O.); andrej.vladimirov.1995@mail.ru (A.V.); sknew@yandex.ru (S.G.K.); anton.chugunov@gmail.com (A.O.C.); shifter2007@gmail.com (Y.A.A.); r-efremov@yandex.ru (R.G.E.); vkublitski@pharmabio.ru (V.S.K.); 2Institute of Molecular Medicine, Sechenov First Moscow State Medical University, 119991 Moscow, Russia; 3National Research University Higher School of Economics, 101000 Moscow, Russia; 4Moscow Institute of Physics and Technology (State University), Dolgoprudny, 141701 Moscow Oblast, Russia; 5Branch of the Shemyakin-Ovchinnikov Institute of Bioorganic Chemistry, Russian Academy of Sciences, 6 Nauki Avenue, 142290 Pushchino, Russia; viktorpalikov@mail.ru (V.A.P.); yuliyapalikova@bibch.ru (Y.A.P.); shaykhutdinova@bibch.ru (E.R.S.); dyachenko@bibch.ru (I.A.D.); 6Federal State Budgetary Institution «Scientific Center of Biomedical Technologies of Federal Medical and Biological Agency» (FSBI SCBT FMBA of Russia), 1 Svetlye Gory, Moscovskaya Oblast, 143442 Krasnogorskiy Rayon, Russia; agvozd33@gmail.com

**Keywords:** acid-sensing ion channel, sevanol, electrophysiology, molecular docking, nociception, analgesia, lignan, total synthesis

## Abstract

Among acid-sensing ion channels (ASICs), ASIC1a and ASIC3 subunits are the most widespread and prevalent in physiological and pathophysiological conditions. They participate in synaptic plasticity, learning and memory, as well as the perception of inflammatory and neurological pain, making these channels attractive pharmacological targets. Sevanol, a natural lignan isolated from *Thymus armeniacus*, inhibits the activity of ASIC1a and ASIC3 isoforms, and has a significant analgesic and anti-inflammatory effect. In this work, we described the efficient chemical synthesis scheme of sevanol and its analogues, which allows us to analyze the structure–activity relationships of the different parts of this molecule. We found that the inhibitory activity of sevanol and its analogues on ASIC1a and ASIC3 channels depends on the number and availability of the carboxyl groups of the molecule. At the structural level, we predicted the presence of a sevanol binding site based on the presence of molecular docking in the central vestibule of the ASIC1a channel. We predicted that this site could also be occupied in part by the FRRF-amide peptide, and the competition assay of sevanol with this peptide confirmed this prediction. The intravenous (i.v.), intranasal (i.n.) and, especially, oral (p.o.) administration of synthetic sevanol in animal models produced significant analgesic and anti-inflammatory effects. Both non-invasive methods of sevanol administration (i.n. and p.o.) showed greater efficacy than the invasive (i.v.) method, thus opening new horizons for medicinal uses of sevanol.

## 1. Introduction

Pathological conditions, such as tissue damage, cancer and inflammation, are often accompanied by the acidification of the extracellular environment, which is a primary cause of pain in the sufferer. Acid-sensing ion channels (ASICs), members of the degenerin/epithelial Na^+^-channels family, are sensitive to even a slight decrease of the extracellular pH [1,2]. In mammals, four genes encode at least six ASIC isoforms: ASIC1a and ASIC3 isoforms are the most studied channels, and they contribute substantially to different physiological processes and pathological conditions. ASIC1a is expressed throughout the central and peripheral nervous systems, and plays an important role in synaptic plasticity, learning and memory, as well as in ischemic processes and anxiety disorders [3,4,5]. ASIC3 is widely distributed in the peripheral nervous system and non-neuronal tissues, and is involved in the perception of acid-mediated inflammatory pain (both acute and chronic) [6,7], as well as pain from various mechanical stimuli [8,9].

As a result, these channels are regarded as promising targets for the development of analgesic and anti-inflammatory drugs. Indeed, ASIC channels were shown to be inhibited for low-molecular weight compounds of various natures and may, therefore, be promising objects for the study of channels and the design of drugs. Amiloride (a potassium-sparing diuretic agent) inhibits all of the functional isoforms of ASICs (IC_50_ of 10–60 µM) [10]. A-317567 non-selectively inhibits the ASIC currents (IC_50_ of 2–30 µM) [11]. Nonsteroidal anti-inflammatory drugs act on different isoforms of ASIC: ibuprofen and flurbiprofen inhibit the ASIC1a-containing channels (IC_50_ of 350 µM), whereas aspirin and diclofenac inhibit the ASIC3-containing channels (IC_50_ of 260 and 92 µM, respectively) [12]. Diarylamidines cause the non-specific inhibition of neuronal ASIC currents (IC_50_ of 0.3–40 µM) [13,14]. NMDA receptor channel blockers, such as memantine, IEM-2117 and 9-aminoacridine, inhibit ASIC1a in submillimolar concentrations [15]. Most of these compounds are applied in clinics, and they have all demonstrated analgesic, anti-inflammatory or neuroprotective effects in tests in vivo [4,7,11]. More effective inhibitory activity on ASIC1a and/or ASIC3 isoforms has been identified in peptide molecules from spider [16,17], snake [18,19] and sea anemone [20,21,22,23] venoms, but such animal toxins have not thus far been successfully rendered as medicinal drugs.

Sevanol, a lignan isolated from *Thymus armeniacus*, inhibited human ASIC3 channels with IC50 of 0.3 mM and possessed pain relief activity [24]. The parenteral administration of sevanol (0.01–10 mg/kg) in mice significantly reversed Complete Freund’s adjuvant (CFA) -induced thermal hyperalgesia and reduced the number of writhes in an acetic acid-induced writhing test [25]. An effective scheme of chemical synthesis for sevanol and the fragment making up half of its ((1*S*,2*S*)-1-{[(2E)-3-(3,4-dihydroxyphenyl)prop-2-enoyl]oxy}propane-1,2,3-tricarboxylic acid, according to IUPAC) was developed, and the biological activity of the synthetic compounds was confirmed in vitro as well as in vivo [26,27]. According to previous work [26], the biological activity of Sevanol’s diastereomer decreased on 20% of the natural Sevanol activity in electrophysiological experiments.

The investigation of structure–activity relationships (SAR) is the one of the effective tools for understanding the ligand–receptor interaction and, consequently, for drug design. Here, we present an improved scheme for the synthesis of sevanol and its analogues, and demonstrate a correlation between the activity of the compounds and the number of free carboxyl groups, as well as other physico-chemical descriptors. The competition of sevanol with the RF-amide peptide for ASIC1a modulation, together with molecular modeling, allowed us to predict the presence of the sevanol molecule’s binding site in the central vestibule of the channel. We also demonstrate for the first time that sevanol is effective as an analgesic and anti-inflammatory agent by non-invasive intranasal and, to a greater extent, oral administration. This opens up promising prospects for its use in medicine.

## 2. Results and Discussion

### 2.1. Chemistry

#### 2.1.1. Sevanol Production

In the present study, we tried to trace the correlation between the structure of the sevanol molecule (**I**) and its biological activity. We modified the carboxylic part of **I** and obtained sevanol derivatives: s788 (**II**), s590 (**III**) and epiphyllic acid EA (**IV**) (Figure 1).

The synthesis of sevanol (**I**) was completed based on the method described previously [27]. The synthetic path included the parallel synthesis of two molecules: the t-butyl isocitrate **1** and the suitably protected caffeic acid **2** (Scheme 1). The method of obtaining the intermediate **4** included removing protecting groups (PG) from the phenolic hydroxyls of **3**. Our careful search for the optimal protecting group has been discussed elsewhere [27].

The first branch of the total synthesis of sevanol (**I**) included the preparation of a suitably protected caffeic acid derivative **2** (Scheme 1). However, even though methoxymethyl acetal (MOM) seemed quite suitable as a protecting group before, the deprotection of caffeic isocitrate ester 3a using a mixture of trifluoroacetic acid (TFA) and H_2_O in ratio 4:1 gave the key intermediate **4** with a moderate yield of 53%. Moreover, in order to produce MOM-protected caffeic acid **2a**, the step of the alkylation of the 3,4-dihydroxybenzaldehyde proved problematic and we encountered issues, such as poor yield due to high impurity formation and the reaction tarring. Therefore, we attempted to improve the yield by changing the protective group to acetyl (Ac), as was described in [28]. Commercially available caffeic acid was carefully treated with five equivalents of Ac_2_O in the presence of ten equivalents of pyridine (Py) at −20 °C. The reaction was stirred over 16 h at room temperature. The resulting product **2b** was isolated using recrystallization as a powder with a 91% yield. The corresponding acid chloride **2c** was obtained using 1.2 eq. of thionyl chloride and toluene as a solvent in the presence of a catalytic amount of dimethylformamide (DMF). The reaction was completed after four hours at 100 °C. Thus, we significantly increased the yield of **2** from 53% to 91% and simplified the purification of the desired product **2**.

The second branch of the total synthesis of sevanol was devoted to the synthesis of the tri-*t*-butyl ester of isocitric acid (**1**). According to our initial work describing the total synthesis of sevanol **I**, the compound **1** was obtained according to the method described [29] with an overall yield of 17%. However, we decided to modify it in order to improve the efficiency of our synthetic pathway of sevanol (**I**) by decreasing the number of stages and increasing the total yield of *t*-butyl isocitrate **1** (Scheme 2). The pathway reported by Calo et al. started from (l)-malic acid (**5**) and contained seven synthetic steps. The initial reaction medium remained the same as published in the original source, but the reaction time increased from 24 h to 72 h at room temperature. As a result, the molecule **6** was obtained with an 83% yield. The interaction of (l)-malic acid (**5**) with compound **6** led to the formation of the bis-*t*-butyl ester of malic acid (**7**) with a 53% yield based on the published method. After decreasing the reaction temperature from room temperature to 0 °C and increasing the 2-*t*-butyl-1,3-diisopropylisourea **6** from three to five equivalents, we found that we could increase the yield of **7** from 53% to 71% on this step. The four steps, following [29], yielded compound **1** with a 17% overall yield. According to data in the literature, unfunctionalized compounds such as allyl bromide, methyl bromide, benzyl bromide and others were used as alkylating reagents for the bis-t-butyl ester of (l)-malic acid **7** [30,31]. Subsequently, these radicals underwent transformation in order to obtain functional groups based on them. In order to reduce the number of steps to afford the target compound **1**, we used the moiety of acetic acid, i.e., *t*-butyl bromoacetate (BrCH_2_COO*t*Bu) as an alkylating reagent for the transformation of molecule **7** in the desired alcohol **1** (Scheme 2). Initially, we attempted to perform the synthesis using two equivalents of LDA and one equivalent of *t*-butyl bromoacetate at −78 °C. However, the yield of the reaction was low (19%), even though the desired product **1** appeared. Therefore, we decided to increase the amount of metal reagent to 2.5 equivalents and the alkylating reagent to 1.5 equivalents. The adjustments made to the synthesis conditions allowed us to double the reaction yield to 53%. Thus, it became possible to produce the *t*-butyl ester of isocitric acid (**1**) in one step. The optimum medium of the last step of the first synthetic approach was the usage of 1.5 equivalents of the alkylating reagent BrCH_2_COO*t*Bu and 2.5 equivalents of BuLi as the metal reagent at −78 °C for 3 h.

As a result, the key intermediate **1** was obtained with a 53% yield on the step. The reduction in the number of steps from seven to three led to a decrease in the loss of intermediates during the intermediate reactions, and accordingly, to an increase in the overall yield of the branch of the synthesis of compound **1** from 17% to 36%.

As mentioned above, the main strategy for the total synthesis of sevanol **I** (Scheme 1) was based on the condensation of two key intermediates **1** and **4**. The acylation of *t*-butyl isocitrate **1** with Ac-protected caffeic acid chloride (**2c**) was performed in dichloromethane in the presence of pyridine as a base, and afforded the compound **3b** with a 84% yield (Scheme 1). The further deacetylation of compound **3b** was carried out with *N*-methylpiperazine in tetrahydrofuran (THF), and led to key intermediate **4**. The yield on the deprotection step was 96%. Thus, we achieved a more optimized route to produce key intermediate **4**. Due to the use of acetyl protecting groups for the phenolic hydroxyls of intermediates **2** and **3**, the overall yield of preparation **4** was significantly increased from 7% to 30% compared to the method we described earlier [27].

The crucial coupling step of compound **4** was carried out according to the procedure we described previously [27]. The optimal reaction medium was carefully selected, and included the dimerization of compound **4** with 2.5 equivalents of iron (III) chloride in acetonitrile in the absence of light at 5 °C for 2 h. After treating the resulting intermediate with 80% aqueous TFA for 1 h at 50 °C, a crude deprotected substance of the target sevanol **I** was identified by ^1^H NMR spectra. Chromatography on a semi-preparative HPLC reverse-phase column using MeCN-H_2_O as eluents, followed by lyophilization, gave sevanol **I** as a mixture of diastereomers in a ratio of 1:3 (isosevanol:sevanol) with a 39% yield.

Summarizing the stage modifications above, we achieved a significant increase in the overall yield of the total synthesis of sevanol (**I**) from 3% [27] to 8%, and reduced the number of synthetic procedures to nine steps.

#### 2.1.2. Synthesis of the Derivatives of Sevanol

The carboxylic acid functional group is an important structural part of many scaffolds used in drugs. The acidity, high water solubility, and ability to establish electrostatic interactions and hydrogen bonds make it highly attractive for its capacity to bind to protein targets [32]. We modified the structure of sevanol **I** by protecting the carboxylic groups with methyl in order to check whether these functional groups were involved in the binding site in ASICs. We produced a methyl ester of sevanol (s788) (**II**) by protecting it with methyl groups directly after the completion of the main synthesis of sevanol **I** according to Scheme 1. The compound I isolated and lyophilized after the last step of the total synthesis of sevanol I described above was dissolved in 4M HCl/MeOH and stored at −20 °C for 72 h. The reaction mixture was carefully evaporated and purified on silica gel to afford the compound **II** (s788) with a 97% yield. The phenolic hydroxyl groups remained unprotected.

We also examined a sevanol derivative by reducing the number of carboxylic acid groups, based on the assumption that the inhibitory effect could weaken due to the reduction of active functional groups. The compound **III** (s590) was obtained according to protocol described above for producing sevanol molecule **I**. However, the condensation of acetyl-protected acid chloride **2c** with (l)-malic acid **7** possessing two *t*-butyl protected carboxylic acid groups, and gave the key intermediate **8** with an 81% yield (Scheme 3). Compound **8** underwent the removal of protective groups using *n*-methylpiperazine, followed by dimerization in the same reaction medium as sevanol (**I**). After treatment with 80% aqueous TFA and purification on a semi-preparative HPLC reverse-phase column, followed by lyophilization, the target sevanol derivative **III** (s590) was obtained with an overall yield of 17%.

The synthesis of compound **IV** (EA) has already been reported in the literature [33]. According to this work, epiphyllic acid (EA) (**IV**) was obtained as a co-product of the condensation reaction of caffeic acid in order to produce americanol and isoamericanol in the presence of horseradish peroxidase (HRP) in a phosphate buffer containing 18% 1,4-dioxane and H_2_O_2_. We performed the synthesis of **IV** (EA) by a new approach based on the procedure developed for sevanol (**I**) production (Scheme 4). Thus, commercially available caffeic acid was transformed into an ethyl ester of caffeic acid **10** in EtOH in the presence of a catalytic amount H_2_SO_4_ under reflux [34]. After the treatment with NaHCO_3_ and evaporation, the desired ethyl caffeate **10** was obtained with an 89% yield. The oxidative coupling of monomer **10** was performed using the conditions described for the preparation of *t*-butyl protected 9,10–diisocitryl ester of epiphyllic acid (sevanol) **I** [27]. Thus, the diethyl ester of epiphyllic acid **11** was produced using 2.5 equivalents of iron (III) chloride in acetonitrile in the absence of light at 5 °C for two hours, with a 37% yield. We tried to deprotect **11** by base hydrolysis with 5M NaOH at room temperature. However, the reaction mixture underwent tarring, and it was almost impossible to isolate the target molecule **IV** (EA). Therefore, we attempted to perform the hydrolysis of **11** in an acidic medium using 4M HCl/dioxane in the presence of H_2_O, followed by purification on a semi-preparative HPLC reverse-phase column and lyophilization. This chosen reaction condition clearly led to the formation of epiphyllic acid **IV** (EA) with a 35% yield on this step.

As a result, a series of sevanol derivatives were designed. All of the compounds, including novel molecules such as **II** (s788) and **III** (s590), were characterized by NMR spectroscopy and high resolution mass spectrometry (HRMS).

### 2.2. ASIC1a and ASIC3 Inhibitory Activity of Sevanol and Its Analogues

Testing the activity of synthetic sevanol (**s706**) (as a mixture of 75% of sevanol and 25% of isosevanol) and its analogues (**s590**, **EA**, and **s788**) was carried out by the standard two-electrode voltage clamp method on X. laevis oocytes expressing rat ASIC1a and ASIC3 channels. The activation of the channels was performed using a rapid change of the solution from conditioning pH 7.4 to a solution with pH 5.5. Sevanol and its two analogues, s590 and EA, inhibited ASIC1a and ASIC3 channels. The effect reached a 100% current block, and was reversible. **s788** did not demonstrate any activity in these channels. The dose–response curve and the value of the half-maximal effective concentrations (IC_50_) for both the ASIC1a and ASIC3 isoforms clearly demonstrated the correlation of the channels’ inhibitory effect with the number of carboxyl groups of the compounds (Figure 2).

We computed the hydrophobic/hydrophilic and electrostatic properties of the studied molecules and their distributions over the molecular surfaces of the compounds. All of them exhibited perfect correlation (≥0.9) with IC_50_ (see Appendix A), suggesting that each carboxyl group truly contributes to the inhibiting activity of the ASICs, and the presence at least two of them was essential. Therefore, **s706**, which has the largest number of carboxyl groups (6), showed a stronger and significantly distinguishable inhibitory effect. This difference in the effectiveness of molecules was more clearly manifested for the ASIC3 channel’s activation by acidification: **s706** showed a greater apparent affinity than **s590** (calculated IC_50_ values 175 ± 18 μM versus 241 ± 25 μM, *p* = 0.016, respectively); **s590**, in turn, showed a greater apparent affinity than epiphyllic acid **EA** (241 ± 25 μM versus 424 ± 33 μM, *p* = 0.002). Compound **II**, **s788**, has as many carboxyl groups as sevanol, but all of them are blocked by methyl groups. Sevanol was 1.3 times more effective in inhibiting ASIC3 than ASIC1a. The same can be observed for **s590**. EA acted equally on the ASIC1a and ASIC3 channels. As a result, it can be argued that sevanol and its two truncated analogs inhibit both subtypes of acid-sensing channels; however, the removal of each pair of free carboxyl groups decreased the activity.

It is known that RF-amide peptides produce a potentiating effect on ASIC1a by inhibiting the channel’s desensitization in the presence of protons (during acidic stimulation). The peptides exhibit this effect both in the case of desensitization following the activation of the channels and in the case of steady-state desensitization, when the channels cease to respond to the acid stimulus bypassing the activation stage [35,36]. Using molecular docking and site-directed mutagenesis, it has been shown that RF-amide peptides are most likely to bind to ASIC1a in the central vestibule of the chicken channel [37]. We examined the inhibitory effect of sevanol on the ASIC1a channel activated by pH 5.5 in the presence of 200 μM FRRFa peptide (Phe-Arg-Arg-Phe-amide), and showed that sevanol and the peptide compete for the binding site. Figure 3A shows that, in the presence of the peptide, the characteristic form of the current with delayed desensitization persists upon co-application with sevanol. The peptide worsened the effectiveness of the inhibitory action of sevanol on the channel by 1.5 times, and reduced the Hill coefficient (Figure 3). Thus, it can be assumed that the binding site of sevanol also lies in the region of the central vestibule.

### 2.3. Molecular Docking

Taking into account the conclusion about the occupation of the central pore, we performed a molecular docking of sevanol and **s788** into a model of the closed rASIC1a, and focused on the central vestibule, virtually excluding the classic acidic pocket from consideration (see Methods). The top 20 sevanol–rASIC1a docking solutions were found in the central vestibule, approximately on the channel’s axis (according to Autodock scoring, the calculated free energy of binding is ΔG _binging_ = −7.01 ± 0.12 kcal/M), not occupying the vestibule’s side cavities (Figure 4). Our electrophysiological experiments show that sevanol and FRRFa compete to a small extent for the binding site, and this fact is confirmed by the docking results. Recently, FRRFa has been predicted to bind in the side cavities of the vestibule, coordinated by three glutamic acid and one valine residues that have been confirmed to be important for this binding [37]. Sevanol, as a highly acidic molecule, is unlikely to occupy the same site due to a presumed strong electrostatic repulsion. Most likely, the sevanol and FRRFa sites in the central vestibule partially overlap (Figure 4B). This was confirmed by the concurrent binding between these compounds in electrophysiology.

The presumed sevanol binding site includes three Arg 369 residues from the adjacent rASIC1a subunits. Sevanol probably forms at least two ionic bridges with these residues through its carboxyl moieties—one from the each ‘branch’ of the carboxyl groups. Molecular dynamics modeling and/or the further determination of the complex structure should reveal if it is possible to simultaneously form three ionic bridges. The molecular flexibility of the sevanol and ASIC1a molecules may permit such binding. Apart from Arg 369, three Lys 372 residues are also present in the binding site (≈5 Å ‘above’ arginines) and may electrostatically interact with sevanol (Figure 4C), although we did not observe this in our docking—sevanol typically neighbors them only by the hydroxyl groups of the polyphenolic part of the molecule. Besides the two ‘rings’ of positively charged residues, there are also three ‘rings’ of negatively charged glutamic acid residues: Glu 416 at the level of Arg 369, and Glu 411 and Glu 373 at the ‘upper level’ of Lys 372. The latter may somehow neutralize Lys 372 and leave only the possibility to interact with sevanol hydroxyl groups, while at the ‘lower level’, Arg 369 and Glu 416 optimally alternate in space, thus permitting sevanol accommodation in the binding site.

The docking of the compound s788 (not shown) reveals the obvious fact that this molecule, although it fits the binding site, cannot form ionic bridges with Arg 369 residues due to the absence of the carboxyl groups. Thus, Arg 369 residue seems to be critical for the binding of sevanol and its analogues and for antagonistic activity, which is the subject of the future mutagenesis validation. It is worth noting that Arg 369 has been confirmed to mediate the antagonistic effect of quercetin by mutagenesis to Ala [38]; therefore, this residue may be crucial for several ASIC1a antagonists.

### 2.4. Analgesic Effects of Sevanol

Sevanol was identified in the acidic extract of *T. armeniacus* as the low molecular weight natural molecule that reversibly inhibited both the transient and the sustained current of ASIC3 channels and produced analgesia [24]. Synthetic sevanol exhibited the same analgesic properties after intravenous (i.v.) and intramuscular (i.m.) administration, as was reported for the natural compound [25]. As demonstrated above, sevanol keeps the best activity among analogues synthesized in this work, and therefore, it is the most promising molecule for further studies. For this reason, it was important to study its effect on animal models using various methods of administration. Both of the above-mentioned methods (i.v. and i.m.) are invasive and limit the potential interest to sevanol as a remedy. The analgesic effect of intranasally (i.n.) and orally (p.o.) administered sevanol was measured and compared with i.v. administration in in vivo models of visceral pain intensity after the intraperitoneal administration of acetic acid and CFA-induced inflammation.

The intraperitoneal administration of acetic acid provokes a constriction response that is considered the definitive measure of visceral pain intensity [39]. Animals were pretreated with synthetic sevanol before the acetic acid injection. The sevanol administration significantly reduced the number of writhes in all of these methods of administration (Figure 5A–C). I.v. and i.n. administration produced a statistically significant effect at doses higher than 0.1 mg/kg, whereas p.o. administration was effective starting from a dose of 0.01 mg/kg. All of the administration routes reduced the number of writhes by more than 50% at a dose of 1 mg/kg. When applied i.n., sevanol could also reach the central nervous system and inhibit ASIC1a, the isoform broadly distributed in the brain, and, for example, indirectly influence the opioid system, resulting in a central analgesia, as was previously described for the ASIC1a inhibitors (PcTx1, lindoldhamine) [40,41]. However, in our experiments, no additional analgesic effect in comparison with i.v. administration was observed. Moreover, the weaker effect of i.n. administration could be the result of the slower distribution of sevanol from the nasal cavities to the peripheral nervous system of the abdomen.

Sevanol showed anti-inflammatory properties in a CFA-induced thermal hyperalgesia test (Figure 5D–E). This effect reached statistical significance at a dose of 1 mg/kg both with i.v. and i.n. administration and, surprisingly, was more pronounced with p.o. administration (the effect was statistically significant at a dose of 0.1 mg/kg). Moreover, a significant reduction in paw oedema was observed only when sevanol was administrated p.o. (Figure 6).

The effectiveness of orally administered sevanol in both tests overcame the effects of the other routes of administration, which may be explained by the partial bioconversion of sevanol during first pass metabolism into a more active metabolite. Most probably, this metabolite, together with sevanol, provided a 10-fold enhanced analgesic and anti-inflammatory effect. Therefore, sevanol can exhibit analgesic and anti-inflammatory properties with high efficiency after non-invasive methods of administration to the organism. That fact makes this molecule attractive for further drug development.

## 3. Materials and Methods 

### 3.1. Experimental General Information 

All of the reactions utilizing moisture-sensitive reagents were performed under an inert atmosphere. All of the commercially obtained reagents were used without further purification. Thin-layer chromatography (TLC) was carried out on pre-coated plates (silica gel 60, F_254_, Fluka), and the spots were visualized with UV and fluorescent lights, or by staining with phosphomolybdic acid stains. Column chromatography was performed on silica gel (0.063–0.2 mm/70–230 mesh ASTM, MN Kieselgel). All of the NMR spectra were obtained on Bruker AVANCE III spectrometers (Bruker BioSpin, Germany) with proton operating frequencies of 300, 600 and 800 MHz. Chemical shifts are reported relative to residue peaks of CDCl_3_ (7.27 ppm for ^1^H and 77.0 ppm for 13C), acetone (2.05 ppm for ^1^H and 29.8 ppm, 206.2 ppm for ^13^C) and D_2_O (4.79 ppm for ^1^H). High-resolution mass spectra (HRMS) were measured on an Agilent 6224 TOF LC/MS System.

### 3.2. Syntheisis of tri-tert-butyl 1-hydroxypropane-1,2,3-tricarboxylate 1

A solution of 2.5 M BuLi in hexane (121.2 mmol, 48 mL) was added dropwise to a solution of *n*,*n*-diisopropylamine (121.2 mmol, 17 mL) in THF (100 mL) under nitrogen at −50 °C. The mixture was stirred for 30 min at −50 °C. After the reaction cooled to −78 °C, a *t*-butyl ester of malic acid 7 (48.7 mmol, 12 g) dissolved in 40 mL THF was added dropwise, and the resulting solution was stirred for 20 min. Subsequently, *t*-butyl bromoacetate (73.1 mmol, 10.6 mL) in 20 mL THF was added dropwise over 10 min at −78 °C. The resulting mixture was stirred for three hours at −20 °C. The progress of the reaction was controlled by TLC using EtOAC/hexane (1:9). After the completion of the reaction, the resulting mixture was quenched with 1M HCl (200 mL), extracted with EtOAc (2 × 70 mL), dried over anhydrous Na_2_SO_4_, concentrated in vacuo and purified by column chromatography eluting with EtOAc/hexane (1:10) to give compound 1 as a colorless oil (9.15 g, 53%): Rf (11%, EtOAc/hexane) 0.33; ^1^H NMR (300 MHz, CDCl_3_) δ_H_: 4.21 (1H, dd, *J* = 5.7, 2.7 Hz), 3.31 (1H, ddd, *J* = 8.6, 6.2, 2.7 Hz), 3.23 (1H, dd, *J* = 27.9, 6.0 Hz), 2.76 (1H, dd, *J* = 16.7, 8.5 Hz), 2.52 (1H, dd, *J* = 16.7, 6.1 Hz), 9.29 (9H, s), 1.47 (18H, s); HRMS: calcld [M + H]^+^ C_18_H_33_O_7_ 361.2220 [M + H]^+^ found 361.2222.

### 3.3. General Procedure for the Preparation of Substituted t-butyl Protected Caffeic Esters 3b, 8

The alcohol (1: 5g, 13.8 mmol; 5: 5g, 20.3 mmol) was dissolved in CH_2_Cl_2_ (50 mL), followed by the addition of pyridine (3b: 18.8 mmol, 1.45 mL; 8: 26.4mmol, 2 mL). The resulting reaction mixture was stirred for 15 min at 0–5 °C in an ice bath. A solution of the corresponding caffeic acid chloride 2c (3b: 4.3 g, 15.2 mmol; 8: 6.3 g, 22.3 mmol) in CH_2_Cl_2_ (20 mL) was added dropwise to the stirred mixture. The resulting solution was stirred under nitrogen for 2 h at room temperature, quenched with 1M HCl (30 mL) and extracted with CH_2_Cl_2_ (2 × 30 mL). The combined organic layers were dried (Na_2_SO_4_) and concentrated in vacuo. The residue was purified on silica gel, eluting with EtOAc/hexane (1:3) to afford the product (3b white powder, 7.1 g, 84%; 8: colorless oil 8 g, 81%).

(E)-tri-tert-butyl 1-((3,4-diacetoxystyryl)oxy)propane-1,2,3-tricarboxylate 3b:Rf (33% EtOAc/hexane) 0.3; ^1^H NMR (600 MHz, CDCl_3_) δ_H_: 7.69 (1H, dd, *J* = 8.4, 1.9 Hz), 7.42 (1H, d, *J* = 1.9 Hz), 7.37 (1H, d, *J* = 1.9 Hz), 7.25 (1H, *J* = 8.4 Hz), 6.44 (1H, d, *J* = 16.0 Hz), 5.30 (1H, d, *J* = 3.4 Hz), 3.44 – 3.48 (1H. m), 2.76 (1H, dd, *J* = 16.7, 9.8 Hz), 2.47 (1H, dd, *J* = 16.8, 5.0 Hz), 2.33 (3H, s), 2.32 (3H, s), 1.51 (9H, s), 1.50 (9H, s), 1.48 (9H, s); ^13^C NMR (600 MHz, CDCl_3_): 170.0, 168.7, 167.5, 167.5, 166.1, 164.9, 143.6, 143.2, 141.9, 132.6, 126.0, 123.5, 122.4, 117.7, 82.4, 81.4, 80.5, 71.8, 43.3, 33.3, 27.5, 20.1; HRMS: calcld [M + H]^+^ for C_31_H_43_O_12_ 607.2749 [M + H]^+^ found 607.2745.

(E)-di-tert-butyl 2-((3,4-diacetoxystyryl)oxy)succinate 8: Rf (33% EtOAc/hexane) 0.3; ^1^H NMR (600 MHz, CDCl_3_) δ_H_: 7.69 (1H, d, *J* = 16.0 Hz), 7.42 (1H, dd, *J* = 8.4, 2.1 Hz), 7.37 (1H, d, *J* = 2.1 Hz), 7.24 (1H, d, *J* = 8.4 Hz), 6.45 (1H, d, *J* = 16.0 Hz), 5.45 (1H, dd, *J* = 7.9, 4.7 Hz), 2.88 – 2.80 (2H, m), 2.32 (3H, s), 2.32 (3H, s), 1.49 (9H, s), 1.47 (9H, s); ^13^C NMR (600 MHz, CDCl_3_): 167.9, 167.5, 167.5, 164.9, 143.4, 143.1, 141.9, 132.64, 126.02, 123.5, 122.3, 117.8, 82.2, 81.1, 68.8, 37.1, 27.5, 27.5, 20.1, 20.1; HRMS: [M + H]^+^ calcld for C_25_H_33_O_10_ 493.2068 [M + H]^+^ found 493.2065.

### 3.4. General Procedure for the Preparation of Substituted Caffeic Esters 4, 9

Acetyl-protected caffeic acid ester (3b: 5g, 8.2 mmol; 8: 5g, 10.2 mmol) was dissolved in THF (50 mL) under nitrogen. *N*-methylpiperazine (4: 2 mL, 18 mmol; 8: 2.5 mL, 22.4 mmol) was added dropwise to the mixture at 0–5 °C. The resulting solution was stirred for two hours at room temperature, quenched with 1M HCl (30 mL) and extracted with EtOAc (30 mL). The combined organic layers were dried (Na_2_SO_4_) and evaporated in vacuo without heating. The residue was purified on silica gel, eluting with EtOAc/hexane (1:2) to give light-yellow oil (4: 4.1 g, 96%; 9: 3.4g 85%).

(E)-tri-tert-butyl 1-((3-(3,4-dihydroxyphenyl)acryloyl)oxy)propane-1,2,3-tricarboxylate 4 [27]:

Rf (33% EtOAc/hexane) 0.4; ^1^H NMR (600 MHz, CDCl_3_) δ_H_: 7.59 (1H, d, *J* = 15.9 Hz), 7.04 (1H, d, *J* = 1.8 Hz), 6.93 – 6.88 (2H, m), 6.22 (1H d, *J* = 15.9 Hz,) 5.36 (1H, d, *J* = 3.3 Hz), 3.54 (1H, ddd, *J* = 9.7, 5.2, 3.4 Hz), 2.82 (1H, dd, *J* = 16.8, 9.7 Hz), 2.53 (1H, dd, *J* = 16.8, 5.1 Hz), 1.57 (9H, s), 1.56 (9H, s), 1.53 (9H, s); HRMS: [M + H]^+^ calcld for C_27_H_39_O_10_ 523. 2537 [M + H]^+^ found 523.2539.

(E)-di-tert-butyl 2-((3,4-dihydroxystyryl)oxy)succinate 9:

Rf (33% EtOAc/hexane) 0.25; ^1^H NMR (600 MHz, CDCl_3_) δ_H_: 7.53 (1H, d, J = 15.9 Hz), 7.00 (1H, d, J = 1.7 Hz), 6.87 (1H, dd, J = 8.2, 1.8 Hz), 6.84 (1H, d, J = 8.2), 6.71 (1H, br s, OH), 6.28 (1H, br s, OH), 6.17 (1H, d, J = 15.9 Hz), 5.45 (1H, dd, J = 6.7, 5.8 Hz), 2.89 – 2.84 (2H, m), 1.52 (9H, s), 1.49 (9H, s); ^13^C NMR (600 MHz, CDCl_3_) δ: 168.5, 165.9, 146.2, 145.9, 143.7, 126.7, 121.9, 114.9, 113.9, 113.4, 82.8, 81.6, 68.5, 37.1, 27.5, 27.5; HRMS: [M + H]+ calcld for C_21_H_29_O_8_ 409.1857 [M + H]+ found 409.1861.

### 3.5. Synthesis of Sevanol I

A solution of cinnamate 4 (4 g, 7.76 mmol) in MeCN (40 mL) was stirred at 5 °C in the absence of light while a solution of FeCl_3_ (3.1 g, 19.2 mmol) in H_2_O (30 mL) was added dropwise. The dark green reaction mixture was stirred for 2 h at 5 °C in the absence of light. The resulting mixture was carefully diluted with 0.1M HCl (50 mL) and extracted with toluene (2 × 35 mL). The combined organic layers were dried (Na_2_SO_4_) and concentrated in vacuo. The residue was purified on silica gel in toluene/EtOAc (4:1) in the presence of 3% AcOH to produce a *t*-butyl protected product as a dark yellow oil. The resulting product was quenched with a 20% solution of TFA in water (TFA – H_2_O = 4:1) at 50 °C over 1 h. After some cooling, most of the solvent evaporated without heating in vacuo, and the mixture was purified with a preparative HPLC reverse-phase column. Sevanol I was isolated from the crude material by RP-HPLC on the polystyrene-based resin LPS-500 (Technosorbent, Moscow, Russia) using a Waters PrepLC 2000 Preparative HPLC System equipped with a Waters 2489 dual-wavelength UV detector. The crude material (4 g), dissolved in water (40 mL), was loaded onto the column (5 × 25 cm), equilibrated in 95% of solvent A (0.1% TFA in water) and 5% of solvent B (acetonitrile) at 70 mL/min flow rate, and then eluted by linear acetonitrile gradient (5% B–35% B for 60 min, 70 mL/min). The elution profile was detected at 254 nm and 280 nm, fractions (100 mL each) covering the sevanol elution zone were collected and HPLC-tested for sevanol purity. The fractions containing >98% (both iso-forms in the summary) pure sevanol I were combined and lyophilized. A light yellow powder of sevanol I, as a mixture of two diastereomers in a ratio 1:3, was obtained (650 mg, 15%): ^1^H NMR (600 MHz, D_2_O) δ_H_: 7.75 (1H, s), 7.01 (1H, s), 6.72 (1H, s), 6.69 (1H, *J* = 8.3 Hz, d), 6.62 (1H, *J* = 2.3 Hz, d), 6.43 (1H, *J* = 8.3, 2.4 Hz, dd), 5.38 (1H, *J* = 3.8 Hz, d), 5.33 (1H, *J* = 3.6 Hz, d), 4.50 (1H, *J* = 2.4 Hz, d), 4.09 (1H, *J* = 2.4 Hz, d), 3.56 (1H, *J* = 9.2, 5.4, 3.8 Hz, ddd), 3.46 – 3.43 (1H, m), 2.78 (1H, *J* = 17.3, 9.2 Hz, dd), 2.62 – 2.58 (1H, m), 2.51 (1H, *J* = 17.3, 9.4 Hz, dd), 2.24 (1H, *J* = 17.3, 5.2 Hz, dd); ^13^C NMR (600 MHz, D_2_O) δ: 175.3, 175.1, 173.9, 173.4, 172.7, 171.8, 166.8, 147.6, 143.9, 143.3, 142.8, 141.6, 134.3, 130.8, 123.8, 119.6, 117.4, 116.6, 116.1, 115.3, 72.4, 46.5, 44.0, 42.9, 42.7; HRMS: [M-H]^−^ calcld for C_30_H_25_O_20_ 705.0939 [M-H]^−^ found 705.0937.

### 3.6. Synthesis of Hexamethyl1,1′-((1-(3,4-dihydroxyphenyl)-6,7-dihydroxy-1,2-dihydronaphthalene-2,3-dicarbonyl)bis(oxy))bis(propane-1,2,3-tricarboxylate) II

Sevanol I (20 mg, 0.028 mmol) was dissolved in 4M HCl/MeOH (1 mL). The resulting solution was stirred at −20 °C for 72 h. The reaction mixture was evaporated, purified on silica gel (EtOAc) and lyophilized to give light-yellow powder II (22 mg, 97%):Rf (33% EtOAc/hexane) 0.4; ^1^H NMR (800 MHz, acetone) δ_H_: 7.69 (1H, s), 6.99 (1H, s), 6.73 (1H, s), 6.70 (1H, d, *J* = 8.1 Hz), 6.51 (1H, d, *J* = 2.2 Hz), 6.45 (1H, dd, *J* = 8.2, 2.2 Hz), 5.43 (1H, d, *J* = 4.1 Hz), 5.32 (1H, d, *J* = 3.9 Hz), 4.49 (1H, d, *J* = 2.0 Hz), 4.02 (1H, d, *J* = 2.0 Hz), 3.70 (3H, s), 3.68 (3H, s), 3.66 (3H, s), 3.64 – 3.62 (9H, m), 3.57 (1H, dt, *J* = 9.3, 4.7 Hz), 3.49 (1H, dt, *J* = 9.1, 4.5 Hz), 2.71 – 2.65 (2H, m), 2.81 (1H, dd, *J* = 17.2, 9.4 Hz), 2.72 – 2.65 (2H, m), 2.50 (1H, dd, *J* = 17.2, 4.9 Hz); ^13^C NMR (800 MHz, CDCl_3_); δ: 171.4, 171.3, 170.5, 170.2, 169.9, 168.1, 167.7, 165.0, 147.9, 139.9, 134.7, 130.3, 123.4, 119.8, 118.6, 116.3, 115.1, 114.5, 78.2, 78.1, 77.9, 71.8, 71.5, 51.8, 46.7, 44.8, 42.9, 42.7, 31.5, 31.3, 29.5; HRMS: [M + H]^+^ calcld for C_36_H_39_O_20_ 791.2029 [M + H]^+^ found 791.2031.

### 3.7. Synthesis of 2,2′-((1-(3,4-dihydroxyphenyl)-6,7-dihydroxy-1,2-dihydronaphthalene-2,3-dicarbonyl)bis(oxy))disuccinic acid III

A solution of t-butyl protected caffeic malate ester 9 (0.5 g, 1.2 mmol) in MeCN (5 mL) was stirred at 5 °C in the absence of light while a solution of FeCl_3_ (0.49 g, 3 mmol) in H_2_O (5 mL) was added dropwise. The reaction mixture, protected from the light, was stirred for 2 h at 5 °C. The resulting mixture was carefully diluted with 0.1M HCl (20 mL) and extracted with toluene (2 × 10 mL). The combined organic layers were dried (Na_2_SO_4_) and concentrated in vacuo. The residue was purified on silica gel in toluene/EtOAc (4:1) in the presence of 4% AcOH to produce a *t*-butyl protected product as a dark yellow oil. The resulting product was quenched with a 20% solution of TFA in water (TFA – H_2_O = 4:1) at 50 °C over 45 min. After some cooling, most of the solvent evaporated without heating in vacuo, and the mixture was purified by a preparative HPLC reverse-phase column. The semi-preparative HPLC system Waters 515 was used for this purpose on a tandem of two columns in size 20 × 250 mm reversed-phase sorbents 11AD2 11 microns and LPS-500 70 microns (LLC ‘Technosorbent’. Russia) with a linear gradient of acetonitrile (mobile phase A: 0.1% TFA in water; mobile phase B: acetonitrile; gradient 10–60% for 60 min at a flow rate of 10 mL/min; the profile registration elution was performed by UV absorbance at 254 and 280 nm). A light yellow powder was obtained after lyophilization (124 mg, 17%): ^1^H NMR (600 MHz, D_2_O) δ_H_: 7.69 (1H, dd, *J* = 5.3, 2.1 Hz), 6.92 (1H, d, *J* = 2.3 Hz), 6.65 (1H, d, *J* = 2.1 Hz), 6.61 (1H, dd, *J* = 23.3, 2.3 Hz), 6.55 (1H, dd, *J* = 6.3, 2.2 Hz), 6.37 (1H, dd, *J* = 8.4, 2.0 Hz), 5.34 (1H, dd, *J* = 7.0, 4.2 Hz), 5.25 (1H, dd, *J* = 7.4, 3.7 Hz), 4.46 – 4.37 (1H, m), 4.03 – 3.94 (1H, m), 2.98 – 2.88 (2H, m), 2.84 – 2.73 (2H, m); ^13^C NMR (600 MHz, D_2_O) δ: 173.4, 172.7, 173.0, 166.9, 147.3, 143.9, 143.2, 142.8, 141.1, 134.8, 134.4, 130.7, 123.6, 119.8, 119.5, 117.3, 116.5, 116.3, 116.1, 115.3, 115.1, 69.4, 69.2, 46.5, 46.4, 44.5, 44.2, 35.7, 35.4; HRMS: [M + H]^+^ calcld for C_26_H_23_O_16_ 591.0980 ([M + H]^+^ found 591.0982.

### 3.8. Synthesis of diethyl 1-(3,4-dihydroxyphenyl)-6,7-dihydroxy-1,2-dihydronaphthalene-2,3-dicarboxylate 11

The ethyl ester of caffeic acid 10 (7.2 mmol, 1.5 g) was dissolved in acetonitrile (15 mL) and cooled at 5 °C. A solution of FeCl_3_ (18 mmol, 2.9 g) in water (30 mL) was carefully added to the resulting solution dropwise at 5 °C in the absence of light. The resulting mixture was stirred for 2 h. Afterwards, the reaction was diluted with water (50 mL) and extracted with toluene (2 × 30 mL). The combined organic layers were dried (Na_2_SO_4_) and evaporated in vacuo. The resulting residue was purified on silica gel eluting with EtOAc/hexane (1:1) to afford dark-yellow powder 11 (0.5 g, 37%): Rf (50% EtOAc/hexane) 0.3; ^1^H NMR (600 MHz, acetone-d6) δ_H_: 7.57 (1H, s), 6.96 (1H, s), 6.71 (1H, d, *J* = 8.1 Hz), 6.63 (1H, s), 6.51 (1H, d, *J* = 2.2 Hz), 6.46 (1H, dd, *J* = 8.2, 2.2 Hz), 4.44 (1H, d, *J* = 3.3 Hz), 4.20 – 4.12 (2H, m), 4.09 – 3.99 (2H, m), 3.89 (1H, d, *J* = 3.2 Hz), 1.26 (3H, t, *J* = 7.1 Hz), 1.13 (3H, t, *J* = 7.1 Hz); ^13^C NMR (600 MHz, acetone-d6) δ: 147.2, 144.7, 144.1, 143.7, 137.1, 135.2, 130.0, 123.9, 122.6, 118.8, 116.1, 115.8, 115.0, 114.6, 60.3, 59.9, 47.6, 45.4, 29.4, 29.3, 29.1, 13.6, 13.5; HRMS: [M + H]^+^ calcld for C_22_H_23_O_8_ 415.1387 [M + H]^+^ found 415.1390.

### 3.9. Synthesis of Epiphyllic Acid IV

The compound 11 (0.2 g, 0.46 mmol) was dissolved in dioxane (20 mL), followed by the addition of 4M HCl (10 mL). The resulting solution was stirred for 16 h under reflux, followed by evaporation in vacuo. The residue was purified with a preparative HPLC reverse-phase column. The semi-preparative HPLC system Waters 515 was used for this purpose on a tandem of two columns in size 20 × 250 mm reversed-phase sorbents 11AD2 11 microns and LPS-500 70 microns (LLC ‘Technosorbent’, Moscow, Russia) with a linear gradient of ethanol (mobile phase A: 0.2% TFA in water; mobile phase B: ethanol; gradient 20–60% for 60 min at a flow rate of 10 mL/min; the profile registration elution was performed by UV absorbance at 254 and 280 nm). A light pink powder was obtained after lyophilization (63 mg, 35%): ^1^H NMR (600 MHz, CDCl_3_) δ_H_: 7.64 (1H, s), 6.94 (1H, s), 6.68 (1H, d, *J* = 8.3 Hz), 6.65 (1H, s), 6.59 (1H, d, *J* = 2.2 Hz), 6.43 (1H, dd, *J* = 8.3, 2.2 Hz), 4.41 (1H, d, *J* = 3.2 Hz), 3.83 (1H, d, *J* = 3.1 Hz); ^13^C NMR (600 MHz, CDCl_3_) δ: 170.0, 146.4, 143.4, 142.7, 142.3, 139.1, 134.8, 130.0, 123.5, 121.4, 119.1, 116.5, 115.9, 115.6, 114.7, 46.7, 44.3; HRMS: [M + H]^+^ calcld for C_18_H_15_O_8_ 359.0761 [M + H]^+^ found 359.0762.

### 3.10. Electrophysiological Study on Xenopus Laevis Oocytes 

Oocytes were harvested from female frogs anaesthetized with tricaine methane sulfonate (MS222) (0.17% solution), and the surgery was performed in an ice bath to avoid heavy bleeding. Defolliculated stage IV and V cells were injected with 2.5 and 10 ng cRNA, synthesized from PCi plasmids containing the rat ASIC1a and rat ASIC3 isoforms, respectively, using the Nanoliter 2000 microinjection system (World Precision Instruments, USA). The injected oocytes were kept for 2–3 days at 17 to 19 °C, and then for up to 5 days at 15 °C in a ND96 medium containing (in mM) 96 NaCl, 2 KCl, 1.8 CaCl_2_, 1 MgCl_2_, and 5 HEPES titrated to pH 7.4 with NaOH supplemented with gentamycin (50 μg/mL). Two-electrode voltage clamp recordings were performed at a holding potential of −50 mV using a GeneClamp 500 amplifier (Axon Instruments), and the data were filtered at 20 Hz and digitized at 100 Hz by an AD converter L780 (LCard, Russia). Microelectrodes were filled with 3 M KCl. The conditioning bath solution was ND96; the pH was adjusted to 7.4. The activating test solution was constructed based on the ND-96 buffer, in which 5 mM HEPES was replaced with 10 mM MES and pH was adjusted to 5.5. A computer-controlled valve system was used to achieve a rapid solution exchange in the recording chamber. A four parameter logistic equation was used for the curve-fitting analysis: F(x) = ((a_1_ − a_2_)/(1 + (x/x_0_)*n*)) + a_2_, where x is the concentration of the compound; F(x) is the response value at a given concentration of the compound; a_1_ is the control response value (fixed at 100%); x_0_ is the IC_50_ value; *n* is the Hill coefficient (slope factor); and a_2_ is the response value at the maximal inhibition (% of control).

### 3.11. Molecular Modeling 

Ligands preparation and structure-activity relationship (SAR): all of the ligands that were synthesized were also constructed in the Maestro modeling environment (Schrödinger, LLC, New York, NY, 2012). For the SAR analysis, we calculated several molecular descriptors based on the solvent-accessible surface area (SASA) and the Molecular Hydrophobicity Potential (MHP) [42] and the Electrostatic Potential (ELP) formalisms (our in-house software IMPULSE was used). The correlation coefficients were calculated between the compound activity (IC50) and the calculated quantities. These were: the number of carboxyl groups; the formal charge; SASA; the parameters of the MHP distribution over the molecular surface: MHPmax, MHPmin, MHPmean, MHPstd (maximal, minimal, mean and standard deviations of the MHP values, respectively); the hydrophobic and hydrophilic surface areas (with MHP > 0.2 and < −0.2, respectively. The MHP values are given in logP units, where *p* is the distribution coefficient for the octanol/water binary mixture); ELP is the distribution over the molecular surface statistics: ELPmax, ELPmin, ELPmean, ELPstd (maximal, minimal, mean and standard deviations of the ELP values, respectively; see Appendix A).

#### 3.11.1. Homology Modeling

Since our experiments were conducted on the rat ASIC1 channels, for further docking studies we had to build homology models of these proteins based on the structures of chicken ASIC1 channels available in the Protein Data Bank (PDB). MODELLER 9.19 [43] was used to produce rASIC1 models in open and closed states, using the PDB structures 4NTW and 5WKU, respectively.

#### 3.11.2. Molecular Docking

The prepared rASIC1 models and ligand structures were used for molecular docking in the AutoDock Vina software [44]. Considering that sevanol and its analogues are acidic molecules that are unlikely to occupy the ‘classic’ acidic pocket, we excluded this area from our investigation of possible docking sites. A 56 Å cubic box centered on the vestibule of rASIC1 was used, which is large enough to allow the sampling of the pore region and several of the surface binding sites. The parameters used were as follows: exhaustiveness = 32; energy_range = 10; num_modes = 20. The binding sites’ mapping (which is visualized in Figure 4B) was performed using the CavityPlus webserver [45].

### 3.12. In Vivo Assay

Specific pathogen-free outbred ICR male mice (6 to 8 weeks old, weighing 29 to 33 g) were obtained from the Animal Breeding Facility of the Branch of the Shemyakin-Ovchinnikov Institute of Bioorganic Chemistry of the Russian Academy of Sciences (Pushchino). The animals were acclimatized for 2 weeks before the experimental procedures, and were kept in two-corridor barrier rooms under a controlled environment: a temperature of 20 to 24 °C, a relative humidity of 30% to 60%, and a 12 h light cycle. The animals were housed in Type 3 standard polycarbonate cages (820 cm2) on bedding (LIGNOCEL BK 8/15, JRS, Germany), with ad libitum access to feed (SSNIFF V1534-300, Spezialdiaeten, GmbH) and filtered tap water. The mouse cages were also supplied with material for environmental enrichment, i.e., Mouse House (Techniplast, Italy). Sevanol or the vehicle was administered intravenously, intranasally and orally (30 min before testing).

#### 3.12.1. Abdominal Constriction Test of Visceral Pain

The mice were divided into separate groups, and sevanol or saline was administrated (30 min before testing) as described above. Acetic acid in saline (0.6%, 10 mL/kg) was injected intraperitoneally. The mice were moved into transparent glass cylinders, and the number of writhes was registered for 15 min.

#### 3.12.2. Complete Freund’s Adjuvant-Induced Thermal Hyperalgesia

The development of the inflammation and thermal hyperalgesia of the paw was induced by the injection of the oil/saline (1:1) CFA emulsion into the dorsal surface of the hind paw of the mice (20 μL/paw) 24 h before the measurement. Saline (20 μL) was injected into the control mice. The inflamed paw withdrawal latencies to thermal stimulation were measured on a hot plate device (Hot Plate Analgesia Meter, Columbus Instruments) with a set temperature of 53 ± 0.1 °C and a cut-off time of 60 s.

The significance of the data was determined by analysis of variance (ANOVA), followed by Tukey’s test. Data are presented as mean ± S.D

#### 3.12.3. Ethics Statement

This study strictly complied with the World Health Organization’s International Guiding Principles for Biomedical Research Involving Animals. The research was carried out in the Association for Assessment and Accreditation of Laboratory Animal Care International AAALAC accredited organization according to the standards of the Guide for Care and Use of Laboratory Animals (8th edition, Institute for Laboratory Research of Animals). All of the experiments were approved by the Institutional Policy on the Use of Laboratory Animals of the Shemyakin-Ovchinnikov Institute of Bioorganic Chemistry Russian Academy of Sciences (Protocol Number 267/2018; date of approval: 28 February 2019) and by the Institutional Animal Care and Use Committee (IACUC) of the Branch of the Shemyakin-Ovchinnikov Institute of Bioorganic Chemistry of the Russian Academy of Sciences (identification code: 688/19; date of approval: 10 January 2019).

## 4. Conclusions

We obtained novel, interesting and important results for the analgesic molecule sevanol. At first, an effective synthesis scheme was developed both for sevanol and its analogues, which allowed us to perform an investigation of structure–activity relationships. Based on this, we demonstrated an important role of carboxyl groups for the inhibitory activity of sevanol on the ASIC1a and ASIC3 channels. The studied compounds could be arranged in the following order by the strength of the action on the channels: s706 (sevanol) > s590 > EA, which correlated well with the number of free carboxyl groups in these molecules. The inactive s788 analog proved the importance of the availability of the carboxyl groups as well. The competition with the RF-amide peptide for sevanol measured in electrophysiological experiments in the whole cell configuration helped us to localize the possible binding site for sevanol in the central vestibule of rASIC1a using molecular docking studies. Sevanol possesses a strong analgesic effect, and could be bioavailable when administered by non-invasive methods, as was shown in two different in vivo tests. The abdominal constriction and hot-plate tests clearly showed the benefits of the oral method of sevanol administration, which undoubtedly should be preferable for the use of sevanol as a medicine.

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
