# Peer review of "Sevanol and Its Analogues: Chemical Synthesis, Biological Effects and Molecular Docking"

_pharmaceuticals, 2020, doi:10.3390/ph13080163_

Round 1
Reviewer 1 Report
In this study, Belozerova et. al. describe a new synthetic method for the preparation of Sevanol and its activity as a ASIC3 inhibitor. Furthermore the in vivo effects of Sevanol with diverse administation routes are shown.
In my opinion this study presents some serious flaws that make me recommend its rejection.
1) The described ASIC3 IC50 for sevanol is relatively high (300 uM). Sevanol looks to me as a very promiscous molecule, therefore I am missing some kind of selectivity panel study or knockout mouse study that demonstated the the in vivo activity observed is really result of ASIC3 inhibition by Sevanol. Furthermore, Sevanol was only obtained a mixture of diastereoisomers.
2) The SAR study shown in the paper is very limited and in the level of activitvies measured insufficient to drive meaningfull conclusions.
All in all, I do not think the study meets the necessary quality for publication.
Author Response
1) The described ASIC3 IC50 for sevanol is relatively high (300 uM). Sevanol looks to me as a very promiscous molecule, therefore I am missing some kind of selectivity panel study or knockout mouse study that demonstated the the in vivo activity observed is really result of ASIC3 inhibition by Sevanol. Furthermore, Sevanol was only obtained a mixture of diastereoisomers.
Indeed, the article is chemically oriented and does not include large-scale studies on the extended biological targets panel or knockout animals experiments. The projects financial support does not allow to planning such experiment in this moment. We will attempt to continue the sevanols’ hot spots research and will looking for a practical Sevanol usage as a diastereoisomers mixture, since it is more economically feasible. In the article (10.1134/S1068162015050106) we showed early that the diastereomers' activity is approximately the same.
2) The SAR study shown in the paper is very limited and in the level of activitvies measured insufficient to drive meaningfull conclusions.
The current goal for investigation - a studying of the of carboxyl groups role in the Sevanol’s activity was completely estimated in our opinion. The conclusion about their contribution to overall activity was done on various experiments base. Syntheses of novel analogues will be done soon.
Reviewer 2 Report
The work presented in this manuscript is important and of good interest. The authors make a total synthesis of a natural product and relate it to its biological activity. The lignan synthesis is very well planned and they make important improvements in yields, it is a good synthesis. They also prepare other analogues of lignan to justify and explain the analgesic and anti-inflammatory activity. The manuscript is correctly written, it is of quality and I have not detected errors, the methods and techniques used are well explained.The laboratory work has been carried out correctly, the results obtained are consistent and well justified.The conclusions are important, they explain the results obtained and the importance of the product.
The authors must correct and replace in the Suporting Information the C-13 NMR spectrum of compound 8, the spectrum shown does not seem to correspond to this product.
Author Response
The authors must correct and replace in the Supporting Information the C-13 NMR spectrum of compound 8, the spectrum shown does not seem to correspond to this product.
We replaced incorrect C-13 NMR spectrum for compound 8
Reviewer 3 Report
This manuscript describes an improved synthesis of the natural lignin sevanol, together with the preparation of some esteres derived from it and its biological study together with docking work. It is an interesting exercise in medicinal chemistry and I recommend its acceptance with a few minor changes.
- In Scheme II, it would seem that treatment of tert-butyl alcohol with CDI affords compound 6 in 83% yield. This must be a mistake, or some additional reagent is missing. Also, this part of the work is not well explained in the text.
- In the text accompanying Scheme III, the precise nature of the modifications in the previously published procedure should be explained.
- The biological studies (e.g. on page 8) have been carried out on a mixture of diastereomers, which is not ideal. However, in their previous synthetic paper published in Tetrahedron, the authors describe the preparation of a similar diastereomeric mixture, but they also claim the HPLC purification of the natural product. Why was this method of purification not used here?
- If binding to Arg-369 is critical, compounds s706 and s590 should not be inhibitors, but Figure 2 shows that they have some activity on ASIC channels. Please comment.
- The good oral absorption of sevanol seems strange, in view of its very high hydrophilicity. Can the authors comment on this point?
Author Response
In Scheme II, it would seem that treatment of tert-butyl alcohol with CDI affords compound 6 in 83% yield. This must be a mistake, or some additional reagent is missing. Also, this part of the work is not well explained in the text.
We replaced the Scheme II for new utilized DIC as written in materials and methods.
In the text accompanying Scheme III, the precise nature of the modifications in the previously published procedure should be explained.
We changed the text accompanying Scheme III
The biological studies (e.g. on page 8) have been carried out on a mixture of diastereomers, which is not ideal. However, in their previous synthetic paper published in Tetrahedron, the authors describe the preparation of a similar diastereomeric mixture, but they also claim the HPLC purification of the natural product. Why was this method of purification not used here?
According to our previous work comparing the biological activity of Sevanol’s diastereomer the mixture (1:3 ratio) have no reliable difference with pure Sevanol. So we did not include an additional purifying method for Sevanol and thereby reduced the compound loss and carry out the synthesis more economical. A fact and reference about measured early activity of diastereomer was included in the introduction.
If binding to Arg-369 is critical, compounds s706 and s590 should not be inhibitors, but Figure 2 shows that they have some activity on ASIC channels. Please comment.
We suspect that the reviewer has been mistaken in the binding site interpretation between the ligands and the channel. Both molecules s706 “main” inhibitor (sevanol) and s590 also possessed inhibitory activity are capable to interacting with Arg 369 via carboxyl groups they have. On the contrary, s788, in which free carboxyl groups are absent has no inhibitory activity since this molecule lost the ability to create a salt bridge with Arg 369. As a result, while Arg 369 are associated with ligands the ion channel should pass less current (inhibition effect on Fig 2).
The good oral absorption of sevanol seems strange, in view of its very high hydrophilicity. Can the authors comment on this point?
Undoubtedly, hydrophobic molecules are considered to better penetrate the blood from the gastrointestinal tract. But there are a lot of factors determining the bioavailability of orally administered drugs. The effect of sevanol on hypersensitivity and acid-induced pain was observed within 30 min, therefore sevenol most likely can be absorbed in the stomach. To be well absorbed in the stomach, a drug should be a weakly acidic small molecule (pKa higher than the pH of stomach acid). Sevanol suits well to such conditions. In the acidic environment, sevanol is non-charged (non-ionized) and could move between adjacent epithelial cells (paracellular transport). Additionally, transcellular absorption can occur through the hydrophilic pathway along the narrow aqueous regions that are associated with polar groups of lipids. Known acidic drugs that are well absorbed in the stomach are aspirin and warfarin.
Round 2
Reviewer 1 Report
I feel that there has been no significant change about my main concerns regarding this manuscript and therefore, I cannot recommend its publication.